# Antimycotic Effects of 11 Essential Oil Components and Their Combinations on 13 Food Spoilage Yeasts and Molds

**DOI:** 10.3390/jof7100872

**Published:** 2021-10-16

**Authors:** Laura Nißl, Florian Westhaeuser, Matthias Noll

**Affiliations:** Department of Applied Sciences, Institute for Bioanalysis, Coburg University of Applied Sciences and Arts, 96450 Coburg, Germany; lauraNissl@web.de (L.N.); florian.westhaeuser@hs-coburg.de (F.W.)

**Keywords:** antifungal, checkerboard method, cinnamaldehyde, essential oil, growth kinetics, fractional inhibitory concentration

## Abstract

Food safety is important to reduce food spoilage microorganisms and foodborne pathogens. However, food safety is challenging, as customers’ demand for natural preservatives is increasing. Essential oils (EOs) and their components (EOCs) are alternative antibacterial and antimycotic food additives. In this study, the minimal inhibitory concentrations (MIC) of 11 different EOCs against 13 food spoilage molds and yeasts were investigated via the microdilution method. Cinnamaldehyde (CA) revealed the lowest MIC for all tested strains and all EOCs (32.81–328.1 µg ml^−1^). However, CA is organoleptic and was therefore combined with other EOCs via the checkerboard method. Overall, 27 out of 91 combinations showed a synergistic effect, and both respective EOC concentrations could be reduced by maintaining MIC. Thereby, the combination with citral or citronellal showed promising results. The concentration-dependent effect of CA was studied in further detail on *Saccharomyces cerevisiae*, with CA causing delayed growth-kinetics and reduced total cell numbers. In addition, flow cytometric measurements combined with live–dead staining indicate the fungicidal effect of CA, due to decreasing total cell numbers and increasing relative amount of propidium iodide-positive cells. In this study, we demonstrated that CA is a potent candidate for the use as a natural preservative against food-relevant mold and yeasts showing fungistatic and fungicidal effects. Therefore, CA and EOC combinations with respective lower EOC concentrations reduce organoleptic reservations, which ease their application in the food industry.

## 1. Introduction

In food manufacturing and storage, the fungal spoilage of products causes severe economic losses, impairment of food quality, reduction in nutrient availability and also impacts food safety, due to the potential presence of mycotoxins [1,2,3]. To increase the shelf-life of food, manifold antifungal, mainly chemical, preservatives are in use. Due to the development of resistances to such additives, and increasing consumer interest in both minimally processed food and reduction in chemical preservatives, the need for naturally derived additives to combat fungal decay is emerging [4,5]. Essential oils (EOs) and their components (EOCs) meet these criteria, which are aromatic and volatile liquids extracted from natural raw material of plants. EOs are complex mixtures containing various individual constituents (ICs), such as terpenes, terpenoids, and aromatic compounds [6]. Often, two or three ICs are major components (20–70%), whereas others are present in trace amounts [7]. EOs have shown to act against food spoilage, due to their antiviral [8], antitoxigenic [9,10], antiparasitic [11,12], insecticidal [13,14], antibacterial [15,16,17] and also antifungal [5,18,19] characteristics. The antimicrobial activities of EOs are predominantly linked to their main components, whereas minor components are attributed to synergistically supporting the activities [17,19,20,21,22]. The classification of a range of essential oils and their EOCs as generally being recognized as safe by the United States Food and Drug Administration ensures their application in food. Approximately 3000 EOs are known so far, and 300 of them are commercially important mainly for the flavors and fragrances market, due to their organoleptic activity [23]. For usage as preservative for food products, this intense aroma is undesired and exceeds consumers’ acceptability [24].

Individual EOCs have some significant advantages over EOs. Due to their chemical stability, and the fact that the majority of pure EOCs do not alter their biological properties. In contrast, the composition and antimicrobial effects of EOs vary, due to the geographical origin of plants, plant components, harvest season and extraction method [25,26,27].

The use of combinations of EOCs cause synergistic, commutative, indifferent or antagonistic effects on the MIC of each employed EOC [28]. Synergistic interactions can be found if the combined effect of EOCs are greater in reducing the MIC than the sum of the individual EOCs. A commutative effect is observed when the combination does not change the effect neither negatively nor positively. Indifference indicates that the effect of the combination of two EOCs is the same as the most potent one used alone. Finally, antagonism is observed when the effect of one or both EOCs is less when they are applied together than when individually applied [29,30]. To differentiate between fungistatic and fungicidal effects, MIC analyses can be combined with flow cytometric analyses [31]. Such flow cytometric analyses are linked with two fluorescent agents SYTO9 or *N’*,*N’*-dimethyl-*N*-[4-[(E)-(3-methyl-1,3-benzothiazol-2-ylidene)methyl]-1-phenylquinolin-1-ium-2-yl]-*N*-propylpropane-1,3-diamine (SYBR) and propidium iodide (PI), which are capable of differentiating the viability of bacteria [32] and fungi [33].

To date, many studies have investigated the MICs of EOCs against spoilage bacteria [34,35,36], but publicly available datasets for yeasts and molds are sparse. However, these eukaryotic microorganisms can cause severe diseases [2] and are frequently found in food products [1,3]. In this study, a variety of food industry relevant EOCs from the group of terpenes (such as citral, citronellal, citronellol, limonene, and linalool) and of aromatic compounds (such as carvacrol, cinnamaldehyde (CA), eugenol, geraniol, and isoeugenol) were employed as a survey of their antimicrobial activities against 13 known food spoilage yeasts and mold strains. Some of these strains have so far not been assessed and the information of the antimycotic efficiency of EOCs can be compared to strains with reported information. On the one hand, this study provides an overview of MICs of EOCs, and FIC indices of EOC combinations with CA. On the other hand, we examine the effect of CA via growth curves and live–dead staining of *S.*
*cerevisiae* via flow cytometric analysis to elucidate its antimycotic effect on a single-cell level. Such high-resolution analyses are needed to investigate the fungistatic effects of sub-lethal EOC concentrations on the growth of *S.*
*cerevisiae*.

## 2. Materials and Methods

### 2.1. Strains, Chemicals and Cultivation

All strains (*Aspergillus niger,* DSM 12634; *Fusarium solani*, DSM 1164; *Penicillium funiculosum*, DSM 10640; *Candida parapsilosis*, DSM 5784; *Debaryomyces hansenii*, DSM 70590; *Dekkera bruxellensis*, DSM 70001; *Hansenula anomala*, DSM 70263; *Issatchenkia orientalis*, DSM 6128; *Kluyveromyces marxianus*, DSM 70073; *Pichia membranifaciens*, DSM 70633; *Saccharomyces cerevisiae*, DSM 70499; *Schizosaccharomyces octosporus,* DSM 70573; *Schizosaccharomyces pombe,* DSM 70576) were obtained from the German Culture Collection (DSMZ, Braunschweig, Germany). Strains were kept in Roti^®^-Store cryo vials (Carl Roth GmbH + Co. KG, Karlsruhe, Germany) for conservation. Each strain was cultivated on yeast plates (YM agar; DSMZ medium 186: 3 g L^−1^ yeast extract, 3 g L^−1^ malt extract, 5 g L^−1^, peptone from soybeans, 10 g L^−1^ glucose, agar 1.5% (*w/v*)) or potato dextrose plates for molds (PD agar; 6.5 g L^−1^ potato infusion, 20 g L^−1^ glucose, agar 1.5% (*w*/*v*)). After 3 to 5 days of incubation, strains were transferred onto new plates for subsequent experiments. EOCs carvacrol (CAR, ≥98.00%), cinnamaldehyde (CA, ≥98.00%), citral (CI, ≥96.00%), citronellal (CLA, ≥85.00%), citronellol (CLO, ≥95.00%), eugenol (EU, ≥98.00%), geraniol (GE, ≥97.00%), isoeugenol (IEU, ≥98.50%), limonene (LM), linalool (LN, ≥97.00%) and vanillin (VA, ≥97.00%) were purchased from Sigma-Aldrich Inc. (St. Louis, MO, USA).

### 2.2. Antifungal Susceptibility Tests

The antifungal assays were performed as described previously by the European Committee on antimicrobial susceptibility testing [37,38] with minor modifications. Fourfold concentrated stock solutions and serial twofold dilutions of sterile-filtered (0.2 µm PTFE, Rotilabo^®^, Carl Roth GmbH + Co. KG) EOCs were prepared in 20% Tween 20 (*v*/*v*) (Carl Roth GmbH + Co. KG) and tested in a concentration range of 0.004 to 8 µg µL^−1^ (*w*/*v* or *v*/*v*). For comparison with previously reported MIC values, concentrations were converted into µg mL^−1^, using the following density values: CAR 0.976 g mL^−1^; CA 1.05 g mL^−1^; CI 0.888 g mL^−1^; CLA 0.857 g mL^−1^; CLO 0.855 g mL^−1^; EU 1.067 g mL^−1^; GA 0.879 g mL^−1^; IEU 1.082 g mL^−1^; LM 0.86 g mL^−1^; and LN 0.87 g mL^−1^. The antifungal drug Amphotericin B (Carl Roth GmbH + Co. KG) was diluted in 20% dimethyl sulfoxide (DMSO, Carl Roth GmbH + Co. KG), sterile-filtered (0.2 µm PTFE, Rotilabo^®^, Carl Roth GmbH + Co. KG) and used as a reference in a concentration range from 0.0157 to 16 µg mL^−1^.

The in vitro MIC values were determined via the broth microdilution method in triplicate in a 96-well microtiter plate. Yeast inoculum suspensions were prepared as described previously [37] and were adjusted to an optical density (OD) of 0.12 to 0.15 at a wavelength of 620 nm (OD_620_) by a spectrophotometer (ScanDrop 250, Analytik Jena AG, Jena, Germany), while mold inoculum was adjusted to a cell number of 1.0 × 10^6^ spores mL^−1^ as described earlier [38]. Each well contained 50 µL EOC dilution, 100 µL of twofold concentrated medium (PD or YM bouillon) and 50 µL yeast or mold inoculum. Microtiter plates for yeast susceptibility tests were incubated at 25 °C for 48 h. As growth of *D. bruxellensis* and the mold strains were slow, those were incubated at 25 °C for 72 h. After incubation, OD_620_ was measured with a microtiter plate reader (FLUOstar Omega, BMG Labtech GmbH, Ortenberg, Germany). Growth was defined by a ΔOD_620_ ≥ 0.2, which was the difference between inoculated EOC and corresponding non-inoculated EOC blank value. Therefore, MIC was defined as the lowest concentration, with a ΔOD_620_ < 0.2.

### 2.3. Synergy Tests via Checkerboard Method

To evaluate the synergistic, commutative, indifferent or antagonistic effect of two EOCs, the checkerboard microdilution method was used as introduced by [39] in three biological replicates (see microplate setup in Appendix A). Each well contained 25 µL of each EOC of respective eightfold concentration, 100 µL of twofold concentrated medium (PD or YM bouillon) and 50 µL yeast or mold inoculum and were incubated as described above. In addition, each EOC combination was prepared with 50 µL water instead of 50 µL inoculum as reference to calculate the respective ΔOD_620_ (see Appendix A). Wells without EOCs but with inoculum were used as positive growth controls, whereas wells without EOCs and inoculum were used as negative controls. The ΔOD_620_ was measured after incubation as described above. Due to the improved checkerboard technique with simultaneous determination of MIC and fractional inhibitory concentration (FIC), we used the calculation and classification of the above described effects and calculation of FICs as described previously [28]. Briefly, for calculations, the lowest FICs of wells with ΔOD_620_ < 0.2 along the turbidity/non-turbidity interface were used [40]. The average of the mean FICs of three biological replicates were used to categorize combinations as synergistic (FIC < 1), commutative (FIC = 1), indifferent (1 < FIC ≤ 2) or antagonistic effect (FIC > 2) [28].

### 2.4. Growth Kinetics

The growth kinetics of *S. cerevisiae* with and without CA was evaluated in a 96-well microtiter plate setup over time with three independent replicates. Each well contained 100 µL of twofold concentrated YM medium, 50 µL of *S. cerevisiae* inoculum (see above) and 50 µL of different CA concentrations (0.125 MIC to 2 MIC). In addition, blanks, positive and negative controls were set as described above. The ΔOD_620_ was calculated each 60 min for a period of 48 h as described above.

### 2.5. Flow Cytometric Analyses

*S. cerevisiae* cells with and without CA together with controls (see above) were additionally investigated in six biological replicates by using SYBR Green I (SYBR; 10,000× concentrated in DMSO; Lonza Group AG, Basel, Switzerland) and propidium iodide (PI; 1 mg mL^−1^; Biotium, Inc., Fremont, CA, USA) staining with subsequent flow cytometric analyses, using a NovoCyte Flow Cytometer (Acea Biosciences Inc., San Diego, CA, USA) as described previously [41]. The analysis and gating of data, displayed in logarithmic scale, were performed using the Novo Express software 1.2 (Acea Biosciences Inc. San Diego, CA, USA).

### 2.6. Statistical Analysis

All FICs were expressed as mean ± SE (n = 3). Statistical analyses were performed to evaluate significant differences from flow cytometric cell counts of stained cells and growth kinetics (see Section 3.3) using one-way analysis of variance (ANOVA) and Tukey post-hoc test using OriginPro 2019 (OriginLab Corporation, Northampton, MA, USA). The significance level was set to *p* ≤ 0.05.

## 3. Results

### 3.1. Cinnamaldehyde Showed Highest MIC

All EOCs were emulsified in 5% Tween 20 (*v*/*v*), which had no negative effect on viability and growth kinetics of each strain. CA caused the lowest MIC, ranging from 32.8 to 262.5 µg mL^−1^, followed by CI (111.0–1776 µg mL^−1^) and CLA (107.1–1714 µg mL^−1^) while the other tested EOCs showed higher MIC values (Table 1). 

*P. funiculosum*, *D. hansenii* and *P. membranifaciens* were most sensitive to EOCs compared to the other tested strains. In contrast, *I. orientalis*, *K. marxianus* and *S. cerevisiae* showed higher MICs, compared to other strains (Table 1).

### 3.2. Synergistic Effects of Cinnamaldehyde with EOCs

To test the synergistic, commutative, indifferent or antagonistic effects, CA was combined via the checkerboard method with each of the seven selected EOCs, which also caused low MIC values (see Table 1). CI and CA as well as CLA and CA showed the lowest FICs in combination for the majority of mold and yeast strains (Table 2). Other EOC combinations revealed less synergistic effects, and the combination CLO and CA showed mainly indifferent effects. Yeast strains *P. membranifaciens*, *S. cerevisiae* and both *Schizosaccharomyces* strains were the most sensitive strains with EOC combinations with CA, while *H. anomala* was the most resistant strain (Table 2).

### 3.3. Growth Kinetics of S. cerevisiae were Impaired by CA

The low MIC of CA prompted us to examine the effect of sub-lethal EOC concentrations on the growth kinetics of *S. cerevisiae*. Increasing concentrations of CA revealed a drastically impaired growth rate of *S. cerevisiae,* even at low concentrations (Figure 1). Cells treated with 0.125 and 0.25 MIC of CA showed a significantly delayed growth curve compared to respective positive control without CA treatment (Figure 1, Table 3). Moreover, this delay in growth kinetics was enhanced with increasing CA concentrations (e.g., 0.5 MIC). In addition, MIC and double MIC of CA inhibited growth, fully confirming a fungistatic effect (see Table 1).

### 3.4. Cinnamaldehyde Showed Fungicidal Effects

To gain more information, the same *S. cerevisiae* cultivations were used for flow cytometric analyses in combination with SYBR and PI staining. While the OD_620_ measurements revealed fungistatic effects on growth kinetics (Figure 1), the flow cytometric analyses discovered additionally a fungicidal effect of CA (Figure 2). The total cell number of *S. cerevisiae* (SYBR positive) decreased with increasing CA (Figure 2a), which is in line with OD_620_ measurements (Figure 1). In turn, the relative number of PI-positive cells was significantly higher with MIC and double MIC of CA (Figure 2b), with a reduced cell number compared to inoculum concentration (Figure 2a), indicating a fungicidal effect of CA.

## 4. Discussion

EOCs are used as flavoring agents, and their antifungal activity was previously reported [42,43,44]. CA was the most potent antifungal EOC of the 11 tested EOCs, with the lowest MICs for all 13 food spoilage yeasts and molds (Table 1), which is in line for *A. niger* and *F. solani* [44] and *D. bruxellensis* [45]. In addition, close relative species from the genera *Aspergillus* [42,46], *Penicillium* [47], *Fusarium* [48], and *Candida* [43,49] revealed also low MIC values if CA or cinnamon plant extract were applied. Moreover, many bacterial genera showed low MICs in the presence of CA [35]. The MICs of EOs were species specific (Table 1), which was also reported previously [18,44,45]. Overall, *P. funiculosum*, *D. hansenii* and *P. membranifaciens* were most sensitive to EOCs, whereas *I. orientalis*, *K. marxianus* and *S. cerevisiae* showed generally higher MICs (Table 1).

However, a direct comparison of individual MIC values with published data is complicated, as it is affected by the inoculum size, growth stage, culture medium, emulsifying agent, pH of the medium, incubation time/temperature and different methods for MIC determination [18,23,50]. MIC values can shift between ± one dilution step in independent replicates, but if the MICs are determined in the same experimental setup (dilution of drugs and microorganisms) for FIC analyses, potential errors of individual MICs are extinguished [28]. As CA was the most potent EOC (Table 1), we assessed the antifungal efficacy of CA combined with other subsequent potent EOCs CI, CLA, CLO, GE, IEU and LN on all tested 13 spoilage yeasts and molds (Table 2). Although the majority of studies used FIC analyses to assess antibacterial combinations of EOCs, very few and recent studies also addressed antifungal FIC analyses. The combination of cinnamon and lemongrass showed synergistic effects (FIC = 0.75) for *Aspergillus flavus* [51], which is lower than our FICs for *A. niger* (Table 2). Another study showed that the combination of cinnamaldehyde and citral was effective against *Penicillium expansum* and caused membrane damages, including a loss of membrane integrity and ergosterol content [52]. Therefore, similar membrane damages can be expected in the synergistic interaction of CA and CI in our investigations of *P. funiculosum* (Table 2). Based on our panel of 13 yeasts and molds, *A. niger*, *F. solani* and *K. marxianus* were the most resistant strains with mainly commutative and indifferent FICs (Table 2). For instance, the presence of EO caused a thinner hyphal diameter and hyphal wall and in higher concentration disruption of plasma membranes and disorganization of mitochondrial structures for *A. niger* [53]. Similarly, EO caused deformation and loss of integrity of the cell wall and in higher concentration disruption of plasma membranes in *F. solani* [54]. Although less information is available for other molds and yeasts of this study, the cell wall and its membranes are the first barrier against the presence of EO, and respective modifications can be the reason to survive higher EO concentrations. Besides the primary lesion of cell membrane, the disruption of cell wall integrity, the impairment of ergosterol biosynthesis and CA^2+^ homeostasis, the accumulation of intracellular reactive oxygen species (ROS), DNA damage and the inhibition of specific enzymes were described as target sites of CA and other EOCs [42,55,56,57]. In turn, *Aspergillus ochraceus* and *Escherichia coli* were capable of converting cinnamaldehyde to cinnamic acid, which is indicative of a resistance mechanism against both EOCs [36,46].

*P. membranifaciens*, *S. cerevisiae*, and both *Schizosaccharomyces* strains were more sensitive to EOC combinations (Table 2), indicating that the cell wall and its membrane as well as potential detoxification mechanisms were less effective, compared to other tested strains. Cell wall and detoxification strategies by antioxidant enzymes of *S. cerevisiae* were also affected by the presence of EO [58,59]. The concentration-dependent effect on the growth of *S. cerevisiae* was so far rarely studied, and our findings revealed that growth was not only significantly delayed in the presence of sub-lethal CA (CA concentrations below MIC) but also the proliferation was significantly reduced (Figure 1 and Figure 2). Similar results of delayed growth curves and reduced offspring were observed for *C. albicans* and *C. tropicalis* with sub-lethal CA concentrations and were linked to plasma membrane ATPase activity and ergosterol biosynthesis [43]. Our results also indicate that the struggle with CA was very resource consuming for *S. cerevisiae,* as fewer resources were present for a fast and high number of offspring (Figure 1, Table 3). Moreover, sub-lethal CA concentrations were fungistatic, and once the MIC was applied, they were fungicidal, as the inoculum concentration was decreased after incubation (Figure 2). Although the quantity of offspring is reduced at sub-lethal CA concentrations, their cell membrane integrity remains intact. *S. cerevisiae* was still able to proliferate once CA was removed in fresh media, although it was incubated beforehand for 48 h with up to 0.5 MIC of CA. However, once 1 MIC was applied, the re-growth of *S. cerevisiae* without CA was not observed, underlining a fungicidal effect. Therefore, already sub-lethal CA concentrations are beneficial to extend the EOC shelter against food spoilage. However, once sub-lethal CA concentrations become low or less bio-available in the food matrix, spoilage microorganisms can use this advantage for growth and potentially mycotoxin production.

The application of CA in films showed low MICs for *D. hansenii* species and relative species of the genera *Candida* and *Penicillium* [60] used in our study, but concentration-dependent effects of EO and EOCs ideally combined with flow cytometric analyses should be carried out to ensure a safe extension of food shelf life.

## 5. Conclusions

Our survey extended the panel of EOCs on 13 food spoilage yeasts and molds and established a fundament of corresponding FIC analyses, as such analyses are, so far, in their infancy. Our dataset showed that combinations of EOCs with cinnamaldehyde (CA), especially against *P. membranifaciens*, *S. cerevisiae* and both *Schizosaccharomyces* strains, were powerful as potential combinations to extend food shelf life. Furthermore, via flow cytometric analyses, this study demonstrated the fungicidal effect of CA. However, each food matrix should be analyzed in detail, as EOC concentrations below MIC delay their growth but fungicidal effects were only observed ≥ 1 MIC.

## Figures and Tables

**Figure 1 jof-07-00872-f001:**
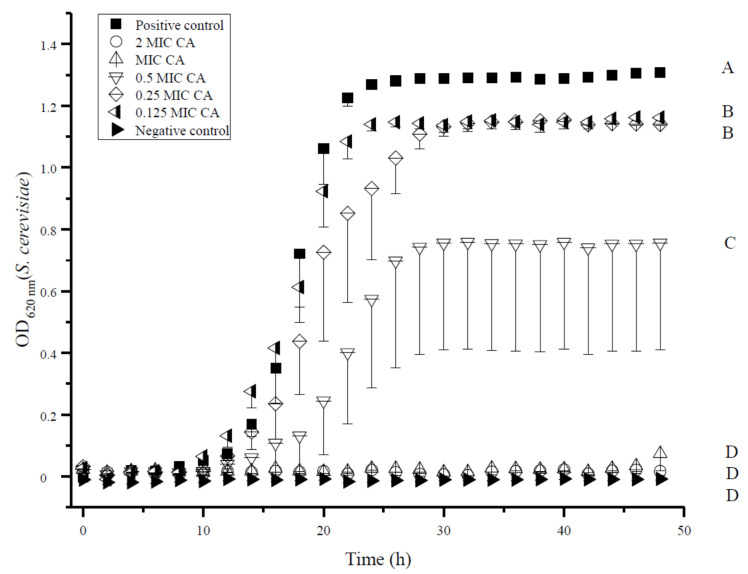
Effect of 0.125 minimal inhibitory concentration (MIC; 0.02 µL mL^−1^) of cinnamaldehyde (CA); 0.25 MIC CA (0.03 µL mL^−1^); 0.5 MIC CA (0.06 µL mL^−1^); 1 MIC CA (0.125 µL mL^−1^); or 2 MIC CA (0.25 µL mL^−1^) compared to negative control without cells and positive control without treatment with CA on the growth of *S. cerevisiae* over time. Error bars (only negative error bars are shown) indicate standard error of independent biological triplicate measurements (*n* = 3). Different letters beside curves indicate significant differences (*p* < 0.05) between different treatments according to one-way analysis of variance.

**Figure 2 jof-07-00872-f002:**
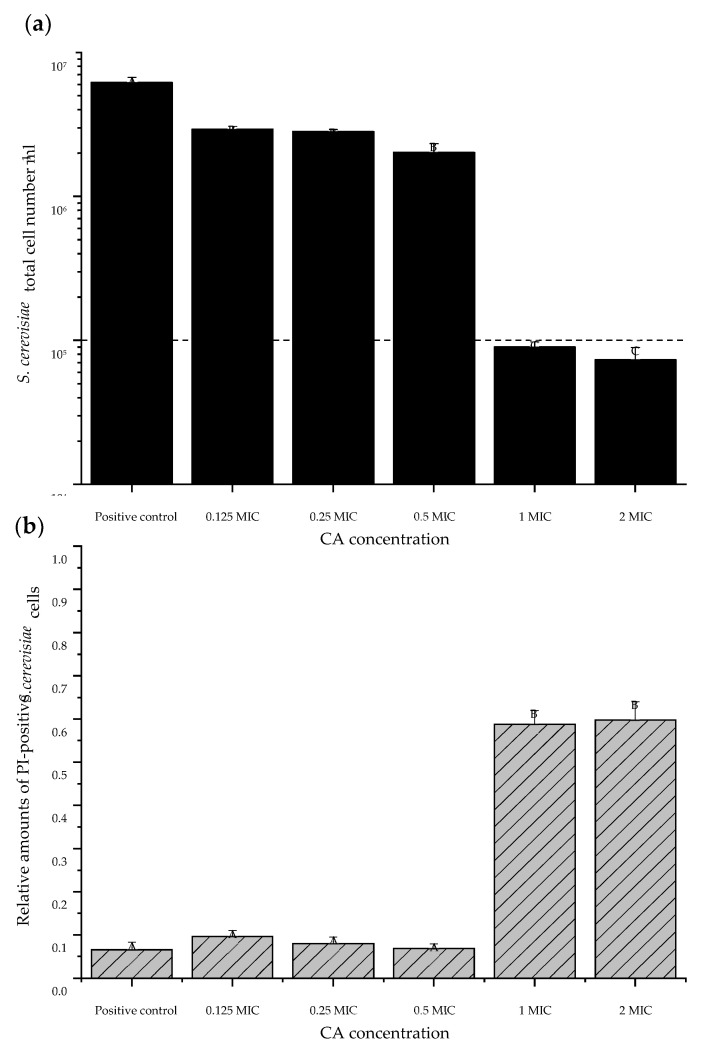
Total cell number (**a**) and relative amount of PI positive of *S. cerevisiae* cells (**b**) after 48 h of growth with cinnamaldehyde (CA). Positive control treated without CA; 0.125 minimal inhibitory concentration (MIC) CA (0.02 µL mL^−1^); 0.25 MIC CA (0.03 µL mL^−1^); 0.5 MIC (0.06 µL mL^−1^) CA; 1 MIC CA (0.125 µL mL^−1^); or 2 MIC CA (0.25 µL mL^−1^). Dashed line (**a**) denotes inoculum concentration. Error bars indicate standard error of six independent replicates (*n* = 6). Different letters above bars indicate significant differences (*p* < 0.05), according to one-way analysis of variance.

**Table 1 jof-07-00872-t001:** Minimal inhibitory concentration (MIC) of tested EOCs against yeast and mold strains as mean of independent biological triplicate measurements (*n* = 3).

EOCs and Control	Amphotericin B	CAR	CA	CI	CLA	CLO	EU	GE	IEU	LM	LN	Vanillin
*A. niger*	4	3904	262.5	888	1714	1710	2134	1758	1082	>3440	1740	1000
*F. solani*	8	3904	262.5	1776	1714	3420	2134	1758	2164	>3440	3480	2000
*P. funiculosum*	>16	3904	131.3	444	428.5	855	1067	879	1082	1720	870	500
*C. parapsilosis*	2	3904	262.5	888	1714	3420	4268	1758	4328	>3440	6960	2000
*D. hansenii*	>4	1952	131.3	444	857	855	2134	879	2164	860	870	1000
*D. bruxellensis*	0.25	1952	32.81	222	107.1	3420	2134	1758	2164	3440	3480	2000
*H. anomala*	0.5	3904	32.81	222	428.5	3420	4268	1758	4328	1720	3480	1000
*I. orientalis*	2	>3904	262.5	888	1714	6840	4268	3516	4328	3440	6960	4000
*K. marxianus*	2	3904	262.5	444	857	3420	4268	1758	4328	3440	3480	4000
*P. membranifaciens*	4	1952	131.3	222	428.5	1710	2134	439.5	2164	1720	1740	1000
*S. cerevisiae*	4	3904	131.3	222	428.5	3420	4268	1758	4328	3440	3480	4000
*S. octosporus*	1	3904	65.63	111	428.5	1710	2134	1758	4328	3440	3480	2000
*S. pombe*	1	3904	328.1	111.5	428.5	3420	2134	1758	4328	3440	3480	2000

Concentration of EOCs were shown as µg mL^−1^. Abbreviations of EOCs are carvacrol (CAR), cinnamaldehyde (CA), citral (CI), citronellal (CLA), citronellol (CLO), eugenol (EU), geraniol (GE), isoeugenol (IEU), limonene (LM), linalool (LN) and vanillin (VA).

**Table 2 jof-07-00872-t002:** FIC indices (dark grey = synergistic; medium grey = commutative; light grey = indifferent) of cinnamaldehyde (CA) in combination with selected EOCs as means ± standard error of independent biological triplicate measurements (*n* = 3).

Mold and Yeast Strains	CA + CI	CA + CLA	CA + CLO	CA + EU	CA + GE	CA+ IEU	CA + LN
*A. niger*	1.0 ± 0.1	1.0 ± 0.1	1.5 ± 0.2	1.3 ± 0.1	1.1 ± 0.1	1.2 ± 0.0	1.1 ± 0.1
*F. solani*	1.0 ± 0.1	1.0 ± 0.0	1.1 ± 0.1	1.1 ± 0.1	1.1 ± 0.1	1.1 ± 0.1	1.0 ± 0.1
*P. funiculosum*	0.9 ± 0.2	1.0 ± 0.1	1.1 ± 0.1	1.1 ± 0.0	1.2 ± 0.2	1.0 ± 0.1	1.1 ± 0.3
*C. parapsilosis*	0.9 ± 0.1	1.0 ± 0.1	1.0 ± 0.0	1.0 ± 0.1	1.1 ± 0.0	1.2 ± 0.1	0.9 ± 0.1
*D. hansenii*	1.1 ± 0.1	1.0 ± 0.1	1.1 ± 0.1	1.2 ± 0.1	1.0 ± 0.1	1.2 ± 0.1	1.2 ± 0.1
*D. bruxellensis*	0.8 ± 0.1	0.9 ± 0.1	1.1 ± 0.0	1.0 ± 0.1	0.9 ± 0.1	1.1 ± 0.1	0.9 ± 0.0
*H. anomala*	1.3 ± 0.1	1.1 ± 0.1	1.5 ± 0.1	1.3 ± 0.2	1.5 ± 0.1	1.5 ± 0.3	1.4 ± 0.3
*I. orientalis*	0.9 ± 0.1	0.8 ± 0.1	1.1 ± 0.1	1.2 ± 0.1	1.0 ± 0.0	1.1 ± 0.1	1.0 ± 0.1
*K. marxianus*	1.0 ± 0.1	1.1 ± 0.1	1.1 ± 0.1	1.1 ± 0.1	1.0 ± 0.1	1.3 ± 0.1	1.1 ± 0.0
*P. membranifaciens*	0.9 ± 0.1	0.6 ± 0.1	1.1 ± 0.1	0.8 ± 0.0	1.1 ± 0.1	0.8 ± 0.0	0.9 ± 0.2
*S. cerevisiae*	1.0 ± 0.1	0.6 ± 0.1	1.1 ± 0.1	0.9 ± 0.1	1.1 ± 0.0	1.0 ± 0.1	1.1 ± 0.0
*S. octosporus*	1.0 ± 0.1	0.9 ± 0.1	0.9 ± 0.1	0.7 ± 0.1	0.7 ± 0.1	0.7 ± 0.1	1.0 ± 0.1
*S. pombe*	0.9 ± 0.1	0.5 ± 0.1	1.0 ± 0.1	0.8 ± 0.1	0.9 ± 0.0	0.9 ± 0.1	0.9 ± 0.1

Abbreviations of EOCs are citral (CI), citronellal (CLA), citronellol (CLO), eugenol (EU), geraniol (GE), isoeugenol (IEU), and linalool (LN).

**Table 3 jof-07-00872-t003:** Significant growth delay of *S. cerevisiae* over time of 0.125 minimal inhibitory concentration (MIC; 0.02 µL mL^−1^) of cinnamaldehyde (CA); 0.25 MIC CA (0.03 µL mL^−1^); 0.5 MIC (0.06 µL mL^−1^) CA; 1 MIC CA (0.125 µL mL^−1^); or 2 MIC CA (0.25 µL mL^−1^) compared to negative control without cells and positive control without treatment with CA in triplicates (*n* = 3). OD_620_ measurements over time (in hours) were carried out and significant differences (*p* < 0.05) between different treatments according to one-way analysis of variance (n.s. = not significantly different after 48 h) denoted.

Treatment	NC	2 MIC	1 MIC	0.5 MIC	0.25 MIC	0.125 MIC	PC
PC	16 h	17 h	17 h	17 h	n.s.	n.s.	
0.125 MIC	13 h	14 h	14 h	19 h	n.s.		
0.25 MIC	19 h	19 h	19 h	n.s.			
0.5 MIC	25 h	25 h	25 h				
1 MIC	n.s.	n.s.					
2 MIC	n.s.						
NC							

## Data Availability

All data are presented in this document and in the Appendix A.

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
