# Peer review of "Antimycotic Effects of 11 Essential Oil Components and Their Combinations on 13 Food Spoilage Yeasts and Molds"

_jof, 2021, doi:10.3390/jof7100872_

Round 1

Reviewer 1 Report

The aim of the paper was to assess the antimycotic effects of 11 essential oil components and their combinations on common food spoilage yeasts and moulds.

It is a very interesting and well-designed study that deserves publication to my opinion.

My only concern, though, is the limited novelty of the work, since several similar investigations are available in the literature.

I suggest that the authors should discuss this point and highlight the innovative elements of their study.

Minor comments

The text should be checked for grammatical and linguistic errors.

Author Response

Reviewer #1

The aim of the paper was to assess the antimycotic effects of 11 essential oil components and their combinations on common food spoilage yeasts and moulds.

It is a very interesting and well-designed study that deserves publication to my opinion.

My only concern, though, is the limited novelty of the work, since several similar investigations are available in the literature.

I suggest that the authors should discuss this point and highlight the innovative elements of their study.

Response: We thank the very positive feedback from reviewer #1. We have re-written the last paragraph of the introduction of the revised manuscript version to point out, why this investigation is of interest of the readership. Moreover, we combined these changes with the requests of reviewer #2.

Minor comments

The text should be checked for grammatical and linguistic errors

Repsonse: We have checked and hopefully we have found all errors in the revised manuscript version.

Reviewer 2 Report

Dear authors,

the work entitled "Antimycotic effects of 11 essential oil components and their combination on 13 food spoilage yeasts and moulds" present significant aspect regarding the subjectof this research. Very interesting is the experimental and results sections regarding the demonstrated synergy of compounds of essential oils. The work was well designed and the different sections were well organized, however  the manuscript requires a further minor revision.

Both in the text and in the references indicated, the references must be introduced in order of year. Correspodingly the order must also be respected in the section "References".

Introduction: 

line 42, page 1: replace references with numbers as in all text.

References:

Recheck all references. Journal names should be abbreviated and written in ITalics

Author Response

Dear authors,

the work entitled "Antimycotic effects of 11 essential oil components and their combination on 13 food spoilage yeasts and moulds" present significant aspect regarding the subjectof this research. Very interesting is the experimental and results sections regarding the demonstrated synergy of compounds of essential oils. The work was well designed and the different sections were well organized, however  the manuscript requires a further minor revision.

Both in the text and in the references indicated, the references must be introduced in order of year. Correspodingly the order must also be respected in the section "References".

Introduction:

line 42, page 1: replace references with numbers as in all text.

Response: We are very thankful for the positive feedback from the reviewer! As already stated by a previous reviewer, we have corrected this error in the revised manuscript version.

References:

Recheck all references. Journal names should be abbreviated and written in ITalics

Response: Thank you again. We have checked all references and made corrections whenever it was needed in the revised manuscript version.

Reviewer 3 Report

The present paper is dedicated to the evaluation of antimycotic activity of several essential oils.

The paper has scientific interest, however improvement of manuscript is necessary.

First of all, both abstract and conclusions are very descriptive and detail results would be useful.

Justify your choice of used essential oils, also indicate the purity of the oils.

42-43 lines - references should be numbered.

Check latin names in all paper - should be in italic.

section 2.3 is not totally clear. for example, "The FIC was calculated as described by [36]", but for briefly explanation, there is citation to ref. [37] (134-136 lines). Also, why such FIC values were selected for interpretation, as in the reference, you used [36], it is "Synergy was defined as an FIC index ≤ 0.5, additivity/indifference was defined as an FIC index > 0.5 to 4, and antagonism was defined as an FIC index > 4".

Check decimal separation of all tables and graph, especially Table 2 and Figure 1.

What was concentration units of amphotericin in Table 1?

It was not clear error bars in Figure 1. Are they only negative? If yes, so why?

Author Response

Reviwer #2

The present paper is dedicated to the evaluation of antimycotic activity of several essential oils. The paper has scientific interest, however improvement of manuscript is necessary. First of all, both abstract and conclusions are very descriptive and detail results would be useful.

Response: We thank reviewer #2 for his supportive feedback. We added additional results details and extended the conclusions in the revised abstract version of the manuscript to meet the request of reviewer #2.

Justify your choice of used essential oils, also indicate the purity of the oils.

Response: We have re-written the last paragraph of the introduction of the revised manuscript version to point out, why these EOCs are of interest. Moreover, we combined these changes with the requests of reviewer #1.

42-43 lines - references should be numbered.

Response: Thank you for this tip. This was an error in our citation software, which has been corrected in the revised manuscript version.

Check latin names in all paper - should be in italic.

Response: We have changed all names and corrected them in the revised manuscript version.

section 2.3 is not totally clear. for example, "The FIC was calculated as described by [36]", but for briefly explanation, there is citation to ref. [37] (134-136 lines). Also, why such FIC values were selected for interpretation, as in the reference, you used [36], it is "Synergy was defined as an FIC index ≤ 0.5, additivity/indifference was defined as an FIC index > 0.5 to 4, and antagonism was defined as an FIC index > 4".

Response: Thank you for your critical reading. We have had a misleading citation (Bonapace et al. 202) here, which we have now updated with the correct citation (Fratini et al. 2017). To improve the readability we have also changed this sentence in the revised manuscript version.  

Check decimal separation of all tables and graph, especially Table 2 and Figure 1.

Response: We have corrected this error of table 2 and figure 1 in the revised manuscript version.

What was concentration units of amphotericin in Table 1?

Response: The concentration units and range from amphotericin B was already named in the Materials and Methods section 2.2 of the first submission. As we do not want to duplicate this information, we have not changed here in the revised manuscript version.

It was not clear error bars in Figure 1. Are they only negative? If yes, so why?

Response: The error bar are valid of course in both directions but to keep the readability of this figure 1 as easy as possible, we show only the negative error bars. We have included in the figure legend 1 of the revised manuscript version additional information to the orientation of the error bars.